# Safety of perioperative treatment with biologics in patients with inflammatory bowel disease undergoing bowel surgery: Experience from a large urban center

**Fabian Schnitzler**[1,2,3]*, **Cornelia Tillack-Schreiber**[1,4], **Daniel Szokodi**[4], **Isabel Braun**[4], **June Tomelden**[5], **Maximilian Sohn**[5], **Franz Bader**[5], **Constanze Waggershauser**[1,4], **Thomas Ochsenkühn**[1,3,4]

1 Department of Medicine II—Grosshadern, Ludwig-Maximilians-University (LMU), Munich, Germany,
2 Praxisklinik München-Pasing, Munich, Germany, 3 Synesis IBD Research Center, Munich, Germany,
4 IBD-Center Munich, Munich, Germany, 5 Department of Surgery, Isarklinikum Munich, Munich, Germany

* fabian.schnitzler@gmail.com

## Abstract

**Data Availability Statement:** All relevant data are within the paper and its Supporting Information files.

### Background and aims

Risks of peri- and postoperative complications after bowel surgery in patients with inflammatory bowel disease (IBD) receiving biologics are still discussed controversially. We therefore addressed the safety of different biologics that were applied in our IBD center before surgery.

### Methods

Data of IBD patients who underwent bowel resections between 2012 and 2022 at our hospital were analyzed retrospectively. Exposure to biologics was defined by receiving biologics within 12 weeks before resective abdominal surgery. Safety considerations included minor complications, such as infections and wound healing disorders and major complications, e.g., anastomotic insufficiency or abscess formation.

### Results

A total of 447 IBD patients (334 with Crohn's disease, 113 with ulcerative colitis), 51.9% female, were included and followed for a median follow-up of 45 months [range 0–113]. A total of 73.9% (326/447) were undergoing medical treatment at date of surgery, 61.5% (275/447) were treated with biologics within 3 months and 42.3% (189/447) within 4 weeks before surgery. Most surgeries (97.1%) were planned electively and 67.8% were performed laparoscopically. Major and minor complications occurred in 20.8% (93/447) of patients. Serious complications were rare: Six patients had acute postoperative bleeding, one CD patient developed peritonitis and two CD patients died postoperatively. After adjusting for age, disease duration, disease activity, Montreal classification, and medical treatment at date of

**Funding:** This work was financially supported by the European Crohn's and Colitis Foundation (ECCS) and HEXAL AG. The funders had no role in study design, data collection and analysis, decision to publish, or preparation of the manuscript.

**Competing interests:** • T. Ochsenkühn has received travel, research and study grants or honoraria for consultations or lectures from AbbVie, Amgen, Biogen, Celltrion, Fresenius, Hexal, Janssen, Lilly, MSD, Viatris, Vifor, Pfizer and Takeda. • F. Schnitzler has received travel grants and honoraria for consultations or lectures from Abbvie, Amgen, Biogen, Hexal, Janssen, Lilly, MSD, Viatris, Pfizer and Takeda. • C. Tillack has received travel grants or honoraria for consultations from Janssen, Galapagos and Lilly. • D. Szokodi has received travel grants or honoraria for consultations or lectures from AbbVie, Amgen, Biogen, BMS, Celltrion, Shire, Lilly, Janssen, Pfizer and Takeda. • M. Sohn has received honoraria for lectures from Galapagos, Lilly and Takeda. • F. Bader has received travel, research and study grants or honoraria for consultations or lectures from Intuitive Surgical Medtronic, MTIGER and Takeda • C. Waggershauser has received travel grands and lecture fees from Biogen, Janssen and Lilly. • This did not alter the authors' adherence to all the PLOS ONE policies on sharing data and materials, as detailed online in the guide for authors. • The other authors have no conflicts of interest to disclose. • No writing assistance was utilized in the production of this manuscript. • There are no patents, products in development or marketed products to declare.

surgery, no significant differences were observed regarding complications and exposure to biologics.

## Conclusions

This retrospective single center study of 447 IBD patients goes to demonstrate that perioperative use of biologics is not associated with a higher risk of complications.

## Introduction

With the introduction of anti-TNF-therapies in the treatment algorithm of inflammatory bowel diseases (IBD) over two decades ago, the era of biologics began and treatment options are still expanding [1–6] with an armamentarium of biologics and small molecules for the treatment of Crohn's disease (CD) and ulcerative colitis (UC). Although it has been demonstrated that in some IBD patients, the course of IBD could be changed with the introduction of biologics resulting in less flares, less hospitalizations and less surgical interventions [7, 8], surgery cannot be avoided in up to 30% of patients with UC and 80% of IBD patients with CD [7, 9–11].

Eventually, many of these are directly exposed to biologics when surgery takes place. The impact of exposure to biologics on surgical outcomes is still discussed controversially, especially because several observational retrospective trials yielded controversial results [12].

There is still concern, that the perioperative use of biologics increases the risk of infectious and wound healing complications like the use of corticosteroids [13–18].

Accordingly, discontinuation of biologics before surgery is still debated and especially how long that time should be [17].

For anti-TNF-treatment, safety of perioperative treatment was addressed in patients with IBD undergoing abdominal surgery in a recently published prospective trial, the "Prospective Cohort of Ulcerative Colitis and Crohn's Disease Patients Undergoing Surgery to Identify Risk Factors for Post-Operative Infection, the PUCCINI trial" [19]. This observational trial demonstrated that exposure to anti-TNF-treatment in these patients 12 weeks before a planned surgery was not associated with increased risk of infectious complications after surgery.

In our actual observational trial, we therefore focused on safety of biologics in patients with IBD from a single center who underwent abdominal surgery and received biologics within 12 weeks before surgery.

## Materials and methods

### Data collection

Clinical data of patients with CD or UC from our registry (MUNICH IBD DATA) were used in this retrospective single center study. This registry is updated with all clinical data by patients and physicians every time patients visit the IBD center Munich (hospital and associated outpatient clinic). Patients of the registry had given written informed consent that their data are collected and can be used retrospectively and in an anonymized mode, according to §15 of the professional regulations for physicians and medical scientists. The registry was established in April 2018 after approval by the ethics committee of the medical association Hamburg, the "Ärztekammer Hamburg" (PV5539, "Long-term observation of IBD patients in a nationwide German IBD Registry "CEDUR"), Hamburg, Germany. In 2020, the Ethics

Committee of the Bayerische Landesärztekammer (No. 2020–1130, MUNICH IBD DATA) Munich, Germany determined in an obligatory and binding fashion, that establishing our own Munich IBD database (MUNICH IBD DATA) does not need an additional consultation with and approval of the Ethics Committee of the Bayerische Landesärztekammer if patients' data were used retrospectively and in an anonymized mode.

In 2022 we decided to retrospectively analyse the data of our registry to examine the safety of the perioperative use of biologics in patients with inflammatory bowel disease.

## Patients

Data of adult patients of the registry who had undergone resective bowel surgery at our IBD center in the department of surgery at the Isarklinikum in Munich, Bavaria, Germany between January 2012, and May 2022, were evaluated and transferred for analysis after anonymization. All those patients had been followed-up before and after abdominal surgery at the IBD center Munich (hospital and outpatient clinic). Data included clinical charts, medical and surgical reports and comprised information on medical treatment, disease location, behavior and activity of UC and CD, pre-operative disease course, planned abdominal surgery or urgent/emergent surgery, perioperative and postoperative outcome, possible treatment associated complications.

For assessment of disease location, the Montreal classification was used [20]. According to the Montreal classification, L1 defines CD of the terminal ileum, L2 CD of the Colon, L3 ileocolonic disease and L4 isolated involvement of the upper GI tract. E1 defines ulcerative proctitis, E2 left-sided CU (distal to splenic flexure), E3 extensive disease (proximal to splenic flexure) [20].

## End points and statistical analysis

Primary outcome was defined as the occurrence of minor and major complications after surgery within a period of 3 months in patients with and without direct exposure to biological treatment in the short-term after surgery.

Minor complications included postoperative development of infections, wound healing complications, the occurrence of nausea, intestinal paralysis, and elevated inflammation parameters. Intestinal paralysis was seen as a prolonged postoperative ileus and is defined as a dysmotility of the small bowel and the colon for more than 72 hours in combination with nausea, and/or vomiting, abdominal pain, and abdominal distension.

Major complications included postoperative development of abscesses at the surgery site, occurrence of anastomotic insufficiency including serious complications defined as acute bleeding complications, development of peritonitis, and death.

Anastomotic insufficiency was seen as anastomotic leakage and defined as overt disruption of the stapler or suture line, identified by endoscopic or surgical exploration.

Statistical analysis was performed with a statistical analysis package from Statistical Analysis System (SAS®) version 9.4 for Windows. Results of the analysis of the quantitative data were presented as either mean ± s. d. and range (Gaussian data) or median and range (non-Gaussian data). Categorical data were summarized as the percentage of the group total. Student's t-test was used for evaluating differences in distributions of quantitative data (Gaussian data) and the Wilcoxon rank-sum test for non-Gaussian data. For investigation of influencing factors, increasing the risk of postoperative complications, logistic regression models were used.

## Results

### Patient cohort and characteristics

The demographic and clinical characteristics of all included patients are described in Table 1. A total of 447 patients with IBD 74.7% (334) with Crohn's disease and 25.3% (113) with ulcerative colitis, including 232 women (51.9%) were admitted at the department of surgery at the Isarklinikum between January 2012 and May 2022. Median age was 44 years [range 19–89], median age at diagnosis was 24 years [range 5–84] and median age at surgery was 41 years [range 16–85] (Table 1). Disease duration until surgery was 11 years [range 0–47], follow-up overall was 45 months [0–113]. Four UC patients and 19 CD patients had their first diagnosis at date of surgery. All patients were followed postoperatively at least for 3 months.

More than half of IBD patients already had undergone disease-related surgery before observed IBD surgery (244/446, 54.7%, Table 1).

Disease activity at date of surgery was documented in 377 of 447 IBD patients (84.3%). With 74.3%, most IBD patients (280/377) state of disease was active at date of surgery with moderate to severe disease activity (Table 1).

In UC patients, exact disease location was documented in 80.5% of patients (91/113). Most UC patients, 71.4% (65/91) suffered from pancolitis; left-sided colitis was seen in 22.0% (20/91) and in 4.4% (4/91) only ulcerative proctitis was observed (Table 1).

In CD, exact disease location was documented in 326/334 (97.6%) patients. Most CD patients had ileocolonic involvement (57.4%, 187/326), in 13.5% of patients (44/326), the colon was affected by CD, 10.4% (34/326) had isolated ileal involvement and in a minority of 2 patients (0.6%) only the upper GI showed CD related alterations (Table 1). In a total of 57 patients (17.5%), the anastomosis was inflamed.

One 57 years old male CD patient with ileocolonic involvement had developed a rectal carcinoma diagnosed at the age of 45 years and had a history of CD for 28 years; he received rectal resection and was never treated with biologics (Table 1).

One further male CD patient developed a neuroendocrine small cell carcinoma in the ascending colon at an age of 54 years with first diagnosis of CD at an age of 22 years. He also never had biologics but had been treated with azathioprine for several years. This patient received right-sided hemicolectomy (Table 1).

Disease behavior is known in 330 of 334 CD patients (98.8%) at date of surgery. With 80.3% (265/330), most CD patients suffered from stricturing disease behavior at date of surgery, 58.2% (104/330) had penetrating disease behavior, whereas a minority of 9.7% (32/330) had non-stricturing and non-penetrating disease behavior at date of planned surgery (Table 1). Overall, 31 patients (9.4%) were diagnosed with abscess formations at date of surgery.

Extra-intestinal manifestations (EIM) were documented in a total of 442/447 IBD patients and occurred only in a minority of 34/442 IBD patients (7.69%, Table 1).

### Perioperative medical treatment

Overall, almost three quarters (73.9%) of all IBD patients (326/447) received medical treatment at date of surgery (Table 1).

61.5% of patients had direct exposure to biologics within 12 weeks before surgery (275/447), and 42.3% (189/447) were exposed to biologics even within 4 weeks prior to surgery.

The majority of those received anti-TNF treatment (36.7% IFX, n = 164, 13.0% ADA, n = 58, 1.1% GOL, n = 6), 29 patients received anti-IL12/23-blocker UST (6.5%), 5.8% were treated with the anti-integrin VDO (n = 26) and two patients had JAK-inhibitor TFO (0.5%).

**Table 1. Patient characteristics.** A total of 447 IBD patients (334 CD, 113 UC, 232 females, 51.9%) were included in this observational retrospective trial. Almost two thirds of all IBD patients (61.52%, n = 275, (214 CD, 61 UC)) had direct exposure to biologic treatment within 12 weeks to CD related surgery. Median follow-up overall was 45 months [range 0–113]. Remission or mild disease activity at date of CD related surgery was documented in only one quarter of IBD patients (25.7%, n = 97/377) whereas 74.3% of all patients (n = 280/377) had moderate to severe disease activity at date of surgery. For description of disease location and disease behavior, the Montreal classification was used [20]. With 73.9% of all patients (n = 326/447), most patients had any treatment at date of surgery with 275 patients receiving biologic treatment (61.5%). A total of 97 patients (21.7%) received corticosteroids at date of surgery. No significant differences were observed between the subgroups with respect to patients' characteristics.

| | | CD | | | UC | | |
|---|---|---|---|---|---|---|---|
| | | Biologics within 12 weeks of surgery | | p-value | Biologics within 12 weeks of surgery | | p-value |
| **Variable** | **Overall (n = 447)** | **Yes (n = 214)** | **No (n = 120)** | | **Yes (n = 61)** | **No (n = 52)** | |
| **Sex (female, %)** | 232 (51.9) | 108 (50.47) | 71 (59.17) | 0.126 | 28 (45.90) | 25 (48.08) | 0.817 |
| **Age (years, median, range)** | 44 [19–89] | 42 [19–89] | 48 [24–88] | **0.009** | 27 [13–77] | 51 [23–82] | **0.009** |
| **Age at diagnosis (years, median, range)** | 24 [5–84] (n = 420) | 23 [5–74] (n = 203) | 24 [8–84] (n = 107) | 0.184 | 27 [13–77] | 27 [10–66] (n = 49) | 0.463 |
| **Age at date of surgery (years, median, range)** | 41 [16–85] (n = 424) | 38 [16–83] (n = 197) | 45 [21–85] (n = 114) | **0.010** | 40 [18–81] | 47 [19–80] | 0.767 |
| **IBD-related surgery before planned surgery (%)** | 244 (54.7) (n = 446) | 146 (68.2) | 70 (58.8) (n = 119) | 0.085 | 10 (16.39) | 18 (34.62) | 0.072 |
| **Disease duration until surgery (years, median, range)** | 11 [0–47] (n = 403) | 11 [0–46] (n = 189) | 13 [0–47] (n = 104) | 0.395 | 9 [0–44] | 12 [0–46] (n = 49] | 0.100 |
| **Follow-up overall (months)** | 45 [0–113] | 43 [0–76] | 42 [0–75] | 0.623 | 62 [6–113] | 52 [6–101] | 0.160 |
| **Disease activity (%)** | | | | | | | |
| **Remission/mild activity (%)** | 97/377 (25.73) | 52/194 (26.80) | 19/94 (20.21) | 0.224 | 12/52 (28.85) | 14/37 (37.84) | 0.131 |
| **Moderate to severe activity (%)** | 280/377 (74.27) | 142/194 (73.20) | 75/94 (79.79) | **0.016** | 40/52 (76.92) | 23/37 (62.16) | 0.131 |
| **Disease location (%)** | | | | | | | |
| **E1, ulcerative proctitis** | 4/91 (4.40) | | | | 0/54 (0.00) | 4/37 (10.81) | 0.056 |
| **E2, Left-sided** | 20/91 (21.98) | | | | 12/54 (22.22) | 8/37 (21.62) | 0.056 |
| **E3, pancolitis** | 65/91 (71.43) | | | | 41/54 (75.93) | 24/37 (64.86) | 0.056 |
| **Others** | 1 patient with pouchitis (1.10), 1 patient with inflammation at the anastomosis (1.10) | | | | 1 patient with inflammation at the anastomosis (1.85) | 1 patient with pouchitis, (2.70) | 0.056 |
| **L1, ileal** | 34/326 (10.43) | 15/210 (7.14) | 19/116 (16.38) | **0.016** | | | |
| **L2, colonic** | 44/326 (13.5) | 29/210 (13.81) | 15/116 (12.93) | **0.016** | | | |
| **L3, ileocolonic** | 187/326 (57.36) | 122/210 (58.10) | 65/116 (56.03) | **0.016** | | | |
| **L4, isolated upper disease** | 2/326 (0.61) | 1/210 (0.48) | 1/116 (0.86) | **0.016** | | | |
| **Anastomosis** | 57/326 (17.48) | 43/210 (20.48) | 14/116 (12.07) | **0.016** | | | |
| **Others** | 1 patient with rectum carcinoma (0.31) 1 patient with colon carcinoma (0.31) | | 1 patient with rectum carcinoma (0.86) 1 patient with colon carcinoma (0.86) | | | | |
| **Disease behaviour (%)** | | | | | | | |
| **B1, non-stricturing, non-penetrating** | 32/330 (9.70) | 13/212 (6.13) | 19/118 (16.10) | **0.003** | | | |

(*Continued*)

**Table 1.** (Continued)

| Variable | Overall (n = 447) | CD Biologics within 12 weeks of surgery Yes (n = 214) | CD Biologics within 12 weeks of surgery No (n = 120) | p-value | UC Biologics within 12 weeks of surgery Yes (n = 61) | UC Biologics within 12 weeks of surgery No (n = 52) | p-value |
|---|---|---|---|---|---|---|---|
| B2, stricturing | 265/330 (80.30) | 177/212 (83.49) | 88/118 (74.58) | 0.051 | | | |
| B3, penetrating | 104/330 (58.18) | 71/212 (33.49) | 33/118 (27.97) | 0.301 | | | |
| Abscess formation | 31/330 (9.39) | 22/212 (10.38) | 9/118 (7.63) | 0.412 | | | |
| **Extraintestinal manifestation (%)** | | | | | | | |
| Skin, erythema nodosum, pyoderma gangraenosum | 13/442 (2.94) | 8/214 (3.74) | 2/119 (1.68) | 0.504 | 2/60 (3.33) | 1/49 (2.04) | 1.000 |
| Eye involvement | 4/442 (0.90) | 2/214 (0.93) | 0/119 (0.00) | 0.539 | 0/60 (0.00) | 0/49 (0.00) | 1.000 |
| Arthralgia | 15/442 (3.39) | 9/214 (4.21) | 3/119 (2.52) | 0.549 | 2/60 (3.33) | 1/49 (2.04) | 1.000 |
| Primary sclerosing cholangitis (PSC) | 2/442 (0.45) | 0/214 (0.00) | 0/119 (0.00) | | 1/60 (1.67) | 1/49 (2.04) | 1.000 |
| **Medical treatment perioperatively (%)** | | | | | | | |
| Medical treatment at date of surgery overall | 326/447 (73.93)) | 214/214 (100.00) | 40/119 (33.61) | <0.001 | 61/61 (100.00) | 15/52 (28.85) | <0.001 |
| Biological treatment within less than 3 months at date of surgery | 275/447 (61.52) | 214/214 (100.00) | 0/120 (0.00) | | 61/61 (100.00) | 0/52 (0.00) | <0.001 |
| Biological treatment within less than 4 weeks of surgery | 189/447 (42.28) | 156/214 (72.9) | 0/120 (0.00) | <0.001 | 33/61 (54.10) | 0/52 (0.00) | <0.001 |
| Infliximab | 164/447 (36.69) | 129/214 (60.28) | 0/120 (0.00) | <0.001 | 31/61 (50.82) | 0/52 (0.00) | <0.001 |
| Adalimumab | 58/447 (12.98) | 47/214 (21.96) | 0/120 (0.00) | <0.001 | 10/61 (12.39) | 0/52 (0.00) | 0.002 |
| Golimumab | 6/447 (1.12) | 1/214 (0.47) | 0/120 (0.00) | 1.000 | 5/61 (8.50) | 0/52 (0.00) | 0.061 |
| Vedolizumab | 26/447 (5.82) | 12/214 (5.61) | 0/120 (0.00) | <0.001 | 12/61 (19.67) | 0/52 (0.00) | <0.001 |
| Ustekinumab | 29/447 (6.49) | 26/214 (12.15) | 0/120 (0.00) | <0.001 | 3/61 (4.92) | 0/52 (0.00) | 0.248 |
| Tofacitinib | 2/447 (0.45) | | | | 2/61 (3.28) | 0/52 (0.00) | 0.499 |
| Only corticosteroid treatment at date of surgery | 38/447 (8.5) | 0/214 (0.00) | 25/120 (20.83) | <0.001 | 0/61 (0.00) | 13/52 (25.00) | <0.001 |
| Only Immunosuppressive treatment at date of surgery (thiopurines/methotrexate) | 9/447 (2.01) | 0/214 (0.00) | 9/120 (7.5) | <0.001 | 0/61 (0.00) | 0/52 (0.00) | |
| Biological treatment + corticosteroids | 52/447 (11.63) | 31/214 (14.02) | 0/120 (0.00) | <0.001 | 20/61 (32.79) | 0/52 (0.00) | <0.001 |
| Biological treatment + immunosuppressive treatment (thiopurines/methotrexate) | 27/447 (6.04) | 17/214 (7.94) | 0/120 (0.00) | <0.001 | 9/113 (7.96) | 0/52 (0.00) | 0.004 |
| Biological treatment + corticosteroids and immunosuppressive treatment (thiopurines/methotrexate) | 7/447 (1.57) | 6/214 (2.80) | 0/120 (0.00) | <0.001 | 1/61 (1.64) | 0/52 (0.00) | 1.000 |

A minority of 8.5% of patients (n = 38) was treated with steroids only, at the time of surgery and 2.0% (n = 9) received immunosuppressive monotherapy with thiopurines or methotrexate.

Combination treatment was also observed in a minority of 6.0% of patients (n = 27), who received immunosuppressants (thiopurines or methotrexate) together with biologics. Notably, 11.6% of patients with biologics also received concomitant steroid treatment (n = 52). And finally, combination treatment with steroids, immunosuppressants and biologics treatment was observed in 1.6% (n = 7, Table 1).

## Performed surgeries

In the majority of the 447 IBD patients, abdominal surgery was planned electively (97.1%, n = 434, Table 2). In only 13 patients (10 CD, 3 UC) surgery had to be planned emergently, as a rescue strategy (2.9%). No significant differences according to timing of surgery were observed with respect to exposure to biologic treatment perioperatively (CD, p = 0.177, UC, p = 0.248, Table 2).

With respect to the surgical approach (open or minimal invasive), abdominal surgery was performed minimally invasive in 67.8% (303/447, Table 2). In 26.4% of patients (118/447), surgery was performed open. In 26 patients (5.8%) surgery had to be switched from a minimal invasive to an open approach, intra-operatively (Table 2). No significant differences according to surgical approach were observed with respect to exposure to biologic treatment perioperatively (CD, p = 0.809, UC, p = 0.09, Table 2).

Bowel resection with no information about the removed segment was documented in 18/447, 4.0% of IBD patients including 13 patients (10 CD, 3 UC) with exposure to biologics perioperatively and 5 patients (4 CD, 1 UC) without biological treatment during surgery (Table 2). Segmental small bowel resection was described in 16.3% of patients (n = 73) including 43 patients (37 CD, 6 UC) exposed to biologic treatment perioperatively and 30 patients (21 CD, 9 UC) without exposure to biologics (Table 2).

An ileocolic resection was performed in 36.2% (n = 162) of all patients (all CD), including 100 patients with exposure to biologics perioperatively and 62 patients without (Table 2).

A proctocolectomy was performed in 15.4% of patients (69/447, all UC) with 43 UC patients exposed to biologics and 26 patients without exposure (Table 2).

In 14.5% of all patients (66/447), a segmental colonic resection was performed, including 36 patients (29 CD, 7 UC) with exposure to biologic therapy and 30 patients (21 CD, 9 UC) without exposure (Table 2).

A resection of the rectum was necessary in 2.2% of all IBD patients (10/447) thereof 4 patients (2 CD, 2 UC) with exposure and 6 patients (no CD, 6 UC) without exposure to biologics perioperatively (Table 2).

Rectum extirpation was necessary in one UC patient (0.2%). This patient had no exposure to biologic treatment perioperatively (Table 2).

Resection of the anastomosis was necessary in 48 CD patients of all IBD patients (10.7%), 36 patients with exposure and 12 patients without exposure to biologics perioperatively (Table 2).

A J-pouch was created in 9.8% (44/447) of all patients (all UC), including 25 patients with exposure to biologic treatment perioperatively and 19 patients with no exposure to biologic treatment (Table 2).

An anastomosis was created in 411/447 IBD patients (92.0%) including 100% of the CD patients and two thirds of the UC patients (Table 2). Overall, 256 patients (214 CD, 42 UC) were exposed to biologic treatment perioperatively and 155 not (120 C, 35 UC, Table 2).

**Table 2. Surgery outcome and complications.** Table 2 shows the surgery outcome in patients overall and in subgroups of patients with exposure to biologic treatment within 3 months of surgery in comparison with surgery outcome in patients without exposure to biologic treatment.

| | | CD | | p-value | UC | | p-value |
|---|---|---|---|---|---|---|---|
| | | Biologics within 12 weeks of surgery | | | Biologics within 12 weeks of surgery | | |
| Variable | Overall (n = 447) | Yes (n = 214) | No (n = 120) | | Yes (n = 61) | No (n = 52) | |
| **Surgical timing (%)** | | | | | | | |
| Elective/staged | 434/447 (97.09) | 210/214 (98.13) | 114/120 (95.00) | 0.177 | 58/61 (95.08) | 52/52 (100.00) | 0.248 |
| Urgent/emergent | 13/447 (2.91) | 4/214 (1.87) | 6/120 (5.00) | 0.177 | 3/61 (4.92) | 0/52 (0.00) | 0.248 |
| **Surgical approach** | | | | | | | |
| Laparoscopic | 303/447 (67.79) | 135/214 (63.08) | 79/120 (65.83) | 0.809 | 50/61 (81.97) | 39/52 (75.00) | 0.09 |
| Open | 118/447 (26.40) | 63/214 (29.44) | 34/120 (28.33) | 0.809 | 8/61 (13.11) | 13/52 (25.00) | 0.09 |
| Convertion, laparoscopic to open | 26/447 (5.82) | 16/214 (7.48) | 7/120 (5.83) | 0.809 | 3/61 (4.92) | 0/52 (0.00) | 0.09 |
| **Surgery (%)** | | | | | | | |
| Bowel resection | 18/447 (4.03) | 10/214 (4.67) | 4/120 (3.33) | 0.400 | 3/61 (4.92) | 1/52 (1.92) | 0.116 |
| Segmental small bowel resection | 73/447 (16.33) | 37/214 (17.29) | 21/120 (17.50) | 0.400 | 5/61 (8.20) | 10/52 (19.23) | 0.116 |
| Ileocolic resection | 162/447 (36.24) | 100/214 (46.73) | 62/120 (51.57) | 0.400 | 0/61 (0.00) | 0/52 (0.00) | 0.116 |
| Proctocolectomy | 69/447 (15.44) | 0/214 (0.00) | 0/120 (0.00) | 0.400 | 44/61 (72.13) | 25/52 (48.08) | 0.116 |
| Segmental colon resection | 65/447 (14.54) | 29/214 (13.55) | 21/120 (17.50) | 0.400 | 7/61 (11.48) | 9/52 (17.31) | 0.116 |
| Resection of anastomosis | 48/447 (10.74) | 36/214 (16.82) | 12/120 (10.00) | 0.400 | 0/61 (0.00) | 0/52 (0.00) | 0.116 |
| Rectum resection | 10/447 (2.24) | 2/214 (0.93) | 0/120 (0.00) | 0.400 | 2/61 (3.28) | 6/52 (11.54) | 0.116 |
| Rectum extirpation | 1/447 (0.22) | 0/214 (0.00) | 0/120 (0.00) | 0.400 | 0/61 (0.00) | 1/52 (1.92) | 0.116 |
| Anastomosis created | 411/447 (91.95) | 214/214 (100.00) | 120/120 (100.00) | 0.400 | 43/61 (70.49) | 34/52 (65.38) | 0.861 |
| J-pouch | 44/447 (9.84) | 0/214 (0.00) | 0/120 (0.00) | 0.400 | 26/61 (42.62) | 18/52 (34.62) | 0.629 |
| Length of resected bowel segment (median, range, cm) | 30 [2–115] (n = 230) | 28 [10–100] (n = 117) | 29 [2–105] (n = 68) | 0.366 | 73 [8–94] (n = 27) | 90 [7–115] (n = 17) | 0.145 |
| **Minor complications** | | | | | | | |
| Impaired wound healing | 36/447 (8.05) | 19/214 (8.88) | 12/120 (10.00) | 0.852 | 2/61 (3.28) | 3/52 (5.77) | 0.912 |
| Infection | 17/447 (3.80) | 11/214 (5.14) | 4/120 (3.33) | 0.852 | 1/61 (1.64) | 1/52 (1.92) | 0.912 |
| Intestinal paralysis | 47/447 (10.51) | 32/214 (14.95) | 9/120 (7.5) | **0.046** | 5/61 (8.20) | 1/52 1.92 | 0.215 |
| Nausea | 16/447 (3.58) | 6/214 (2.80) | 1/120 (0.83) | 0.429 | 3/61 (4.92) | 6/52 (11.54) | 0.297 |
| Elevated inflammatory parameters | 41/447 (9.17) | 20/214 (9.35) | 9/120 (7.50) | 0.565 | 3/61 (4.92) | 9/52 (17.31) | **0.033** |
| Any minor complication | 157/447 (35.12) | 88/214 (41.12) | 35/120 (29.17) | **0.034** | 14/61 (22.95) | 20/52 (38.46) | 0.100 |
| **Major complications** | | | | | | | |
| Abscess formation | 8/447 (1.79) | 3/214 (1.40) | 2/120 (1.67) | 0.851 | 2/61 (3.28) | 1/52 (1.92) | 0.856 |
| Suture insufficiency | 23/447 (5.15) | 12/214 (5.61) | 5/120 (4.17) | 0.851 | 4/61 (6.56) | 2/52 (3.85) | 0.856 |
| Acute bleeding complications | 7/447 (1.57) | 3/214 (1.4) | 4/120 (3.33) | 0.585 | 0/61 (0.00) | 0/52 (0.00) | |
| Peritonitis | 1/447 (0.22) | 1/214 (0.47) | 0/120 (0.00) | 0.585 | 0/61 (0.00) | 0/52 (0.00) | |
| Death | 2/447 (0.45) | 1/214 (0.47) | 1/120 0.83 | 0.585 | 0/61 (0.00) | 0/52 (0.00) | |
| Any major complication | 41/447 (9.17) | 20/214 (9.35) | 12/120 (10.00) | 0.848 | 6/61 (9.84) | 3/52 (5.77) | 0.503 |
| Complications within 10 days after surgery | 72/447 (16.11) | 41/214 (19.16) | 18/120 (15.00) | 0.339 | 10/61 (16.39) | 3/52 (5.77) | 0.078 |
| Complications within 3 months after surgery | 23/447 (5.15) | 11/214 (5.14) | 6/120 (5.00) | 0.955 | 1/61 (1.64) | 5/52 (9.62) | 0.093 |
| Surgery necessary because of complications | 40/447 (8.95) | 19/214 (8.88) | 11/120 (9.17) | 0.814 | 8/61 (13.11) | 2/52 (3.85) | 0.105 |
| Within 10 days after complication | 14/447 (3.13) | 5/214 (2,34) | 2/120 (1,67) | 1.000 | 6/61 (9.84) | 1/52 (1.92) | 0.122 |
| Within 3 months after complication | 13/447 (2.91) | 9/214 (4.21) | 1/120 (0.83) | 0.102 | 2/61 (3.28) | 1/52 (1.92) | 1.000 |

According to performed surgeries, no significant differences were observed in the subgroup of patients whether they were exposed to biological treatment perioperatively or not (p = 0.400 for CD and p = 0.116 for UC, Table 2).

### Postoperative outcomes according to severity of complications

**Minor complications.** Impaired wound healing complications occurred in 8.1% of patients (36/447) with no significant differences in exposed versus non-exposed CD and UC patients to biologic treatment perioperatively (CD, p = 0.852, UC, p = 0.912, Table 2).

Overall, 17/447 patients had infectious complications after surgery (3.8%). Again, no significant differences between the groups were observed (CD, p = 0.852, exposed versus unexposed, UC, p = 0.912 exposed versus unexposed, Table 2).

In CD, more patients with biologics within 12 weeks before surgery reported intestinal paralysis (15.0%, 32/214) as compared to patients without biologics (7.5%, n = 9/120, p = 0.046, Table 2), however, in logistic regression analysis, this effect was not significant (see below and Table 3A). In UC, no significant difference was observed between these groups (p = 0.215, Table 2).

Nausea was reported in a total of 16/447 IBD patients (3.6%) with no significant difference between biological exposed versus unexposed IBD patients (p = 0.429 for CD and p = 0.297 for UC, Table 2).

In UC, more patients without biologics showed elevated inflammatory parameters postoperatively (17.3%, 9/52 versus 4.9%, 3/61, p = 0.033, Table 2), however, in logistic regression analysis, this effect was not significant (see below and Table 3D). In CD, no differences were seen (p = 0.565, Table 2).

**Major complications.** Major complications included postoperative occurrence of abscess formations and anastomotic insufficiency, acute bleeding complications, peritonitis, and death due to complications.

In only eight of all 447 IBD patients (1.8%, 5 CD, 3 UC) abscesses were observed postoperatively with no significant differences between exposed and unexposed patients within 12 weeks. (CD, p = 0.851 versus unexposed; UC, p = 0.856 versus unexposed, Table 2).

Anastomotic insufficiency was observed in a total of 23 IBD patients (5.2%, 17 CD, 6 UC) with no significant differences between exposed and unexposed patients (CD, p = 0.851 versus unexposed, UC, p = 0.856 versus un-exposed, Table 2). No significant difference was observed in CD patients versus UC patients regarding anastomotic insufficiency (CD, 17/334 patients, 5.09%, vs. UC, 6/113 patients, 5.31%, p = 1.000, Table 2).

Acute bleeding complications at the anastomosis occurred in a total of seven CD patients (1.6%), including four patients with no exposure to biologic therapy (4/120, 3.3%, Table 2) and two patients with exposure to adalimumab (2/156, 1.3%) and one patient with exposure to ustekinumab (1/58, 1.7%, p = 0.585, Table 2).

No serious complications were observed in UC patients (Table 2).

One 54 years old CD patient with exposure to infliximab within 4 weeks before small bowel surgery with an open surgical approach developed peritonitis within days after surgery.

Two CD patients died in the postoperative course (Table 2): One 52-years old unexposed female with penetrating and stricturing disease course underwent surgery due to a stenosis in the colon with an open surgical approach. She had received only steroids perioperatively and already had had CD-related surgery in the past. She had come to the hospital with signs of sub-ileus and abdominal pain. Postoperatively minor and major complications with infection at the surgery site and anastomotic insufficiency occurred. Eventually, she needed surgery again because of the complications. In the further postoperative course, she developed peritonitis and died due to infectious complications.

Another 39-years old exposed female CD patient with penetrating disease course and entero-enteral fistula underwent laparoscopic surgery because of fistulae. She was diagnosed with CD at an age of 18 years and had infliximab-treatment four weeks before planned surgery.

**Table 3. A-E.** Multivariate logistic regression on the development of post-operative complications, included the following parameters: Biologics within 12 weeks of surgery, age, disease duration, disease activity (remission/mild and moderate/severe), disease behaviour (Montreal classification), corticosteroid treatment perioperatively, including concomitant steroid treatment.

A

| Variable | Estimate | Standard Error | Odds Ratio | 95% Confidence Interval lower limit | upper limit | p |
|---|---|---|---|---|---|---|
| Intercept | -2.522 | 1.136 | | | | |
| Biologics within 12 weeks of surgery* | 0.691 | 0.398 | 1.995 | 0.914 | 4.354 | 0.083 |
| Age | 0.027 | 0.013 | 1.027 | 1.002 | 1.052 | 0.034 |
| Disease duration until surgery | 0.006 | 0.016 | 1.006 | 0.975 | 1.037 | 0.725 |
| Disease activity** | -0.894 | 0.371 | 0.409 | 0.197 | 0.847 | 0.016 |
| Disease behaviour B1* | 0.881 | 0.598 | 2.413 | 0.747 | 7.789 | 0.141 |
| Disease behaviour B2* | 0.411 | 0.389 | 1.508 | 0.703 | 3.236 | 0.291 |
| Disease behaviour B3* | 0.99 | 0.557 | 2.692 | 0.904 | 8.013 | 0.075 |
| Corticosteroids* | -0.236 | 0.462 | 0.79 | 0.32 | 1.953 | 0.61 |

B

| Variable | Estimate | Standard Error | Odds Ratio | 95% Confidence Interval lower limit | upper limit | p |
|---|---|---|---|---|---|---|
| Intercept | -3.407 | 3.359 | | | | |
| Biologics within 12 weeks of surgery* | 0.553 | 1.157 | 1.738 | 0.18 | 16.796 | 0.633 |
| Age | -0.03 | 0.046 | 0.971 | 0.887 | 1.063 | 0.522 |
| Disease duration until surgery | 0.014 | 0.054 | 1.014 | 0.911 | 1.128 | 0.801 |
| Disease activity** | 0.07 | 1.2 | 1.073 | 0.102 | 11.26 | 0.953 |
| Disease behaviour B1* | -0.096 | 1.221 | 0.908 | 0.083 | 9.937 | 0.937 |
| Disease behaviour B2* | 0.082 | 1.116 | 1.085 | 0.122 | 9.674 | 0.942 |
| Disease behaviour B3* | 0.67 | 1.393 | 1.954 | 0.127 | 29.988 | 0.631 |
| Corticosteroids* | 0.239 | 1.219 | 1.27 | 0.117 | 13.842 | 0.844 |

C

| Variable | Estimate | Standard Error | Odds Ratio | 95% Confidence Interval lower limit | upper limit | P |
|---|---|---|---|---|---|---|
| Intercept | -1.02 | 1.735 | | | | |
| Biologics within 12 weeks of surgery* | 1.1 | 0.681 | 3.003 | 0.79 | 11.407 | 0.106 |
| Age | -0.018 | 0.024 | 0.983 | 0.938 | 1.03 | 0.464 |
| Disease duration until surgery | 0.061 | 0.027 | 1.063 | 1.009 | 1.12 | 0.021 |
| Disease activity** | -0.7 | 0.575 | 0.497 | 0.161 | 1.532 | 0.223 |
| Disease behaviour B1* | -1.242 | 0.624 | 0.289 | 0.085 | 0.981 | 0.047 |
| Disease behaviour B2* | -0.785 | 0.68 | 0.456 | 0.12 | 1.729 | 0.248 |
| Disease behaviour B3* | -0.546 | 1.195 | 0.579 | 0.056 | 6.026 | 0.648 |
| Corticosteroids* | -0.15 | 0.723 | 0.861 | 0.209 | 3.549 | 0.836 |

D

| Variable | Estimate | Standard Error | Odds Ratio | 95% Confidence Interval lower limit | upper limit | p |
|---|---|---|---|---|---|---|
| Intercept | -0.668 | 1.683 | | | | |
| Biologics within 12 weeks of surgery* | -1.026 | 0.645 | 0.358 | 0.101 | 1.268 | 0.111 |
| Age | -0.016 | 0.024 | 0.984 | 0.939 | 1.031 | 0.495 |
| Disease duration until surgery | 0.043 | 0.035 | 1.044 | 0.975 | 1.117 | 0.221 |
| Disease activity** | -0.437 | 0.659 | 0.646 | 0.177 | 2.352 | 0.507 |
| Montreal E1*** | -0.455 | 0.894 | 0.581 | 0.042 | 8.076 | 0.686 |
| Montreal E2*** | 0.367 | 0.65 | 1.321 | 0.265 | 6.59 | 0.734 |
| Corticosteroids* | 1.309 | 0.664 | 3.703 | 1.008 | 13.6 | 0.049 |

E

(*Continued*)

**Table 3.** (Continued)

| Variable | Estimate | Standard Error | Odds Ratio | 95% Confidence Interval lower limit | upper limit | p |
|---|---|---|---|---|---|---|
| Intercept | -0.229 | 2.156 | | | | |
| Biologics within 12 weeks of surgery* | 0.947 | 0.879 | 2.578 | 0.46 | 14.436 | 0.281 |
| Age | -0.034 | 0.032 | 0.967 | 0.908 | 1.029 | 0.289 |
| Disease duration until surgery | 0.031 | 0.046 | 1.032 | 0.944 | 1.129 | 0.492 |
| Disease activity** | -0.668 | 0.786 | 0.513 | 0.11 | 2.392 | 0.395 |
| Montreal E1*** | 0.885 | 1.008 | 4.712 | 0.241 | 92.196 | 0.307 |
| Montreal E2*** | -0.22 | 0.725 | 1.562 | 0.262 | 9.315 | 0.625 |
| Corticosteroids* | 0.956 | 0.752 | 2.601 | 0.596 | 11.349 | 0.204 |

**A.** Multivariate logistic regression model for CD, for minor complications, including intestinal paralysis, nausea, and elevated inflammatory parameters. For parameters including infections at surgery site and wound healing complications, the number of patients in the subsequent cohorts were too small to perform logistic regression models. Only age and disease activity were associated with the occurrence of minor complications (p = 0.034 and p = 0.016, respectively).
**B.** Multivariate logistic regression model for CD, major complications, including acute bleeding complications, peritonitis, and death. None of the comparisons were significant.
**C.** Multivariate logistic regression model for CD, for major complications, including abscess formation and anastomotic insufficiency. Interestingly, non-stricturing and non-penetrating disease behaviour (B1 after Montreal classification) and disease duration until surgery were significantly associated with the occurrence of major complications (p = 0.047 and p = 0.021, respectively).
**D.** Multivariate logistic regression model for UC, for minor complications, including intestinal paralysis, nausea, and elevated inflammatory parameters. Steroid treatment was associated with the occurrence of minor complications (p = 0.049). For parameters, including infections and wound healing complications, the number of patients in the subsequent cohorts were too small to perform logistic regression models.
**E.** Multivariate logistic regression model for UC, for major complications, including abscess formation, suture insufficiency or both. None of the comparisons were significant. Major complications, including acute bleeding complications, development of peritonitis and death were not observed in UC patients (Table 2).

She developed a postoperative disease course with minor and major complications within 10 days after surgery with infection at the surgery site and anastomotic insufficiency. She needed yet another surgery because of complications. After 54 days in the hospital, she died due to infectious complications.

**Postoperative outcome according to time of complications.** In 72 patients (16.1%, 59 CD, 13 UC), complications occurred within 10 days of surgery and in 23 IBD patients (5.2%, 17 CD, 6 UC) later than 10 days after surgery (Table 2) and no significant differences were observed within these two periods regarding complications between the exposed and the unexposed patients:

Within 10 days after surgery: CD: 19.2%, 41/214, versus 15.0%, 18/120, p = 0.339. UC: 16.4%, 10/61, versus 5.8%, 3/52, p = 0.078.

Later than 10 days after surgery: CD: 5.1%, 11/214, versus 5.0%, 6/120, p = 0.955. UC: 1.6%, 1/61, versus 9.6%, 5/52, p = 0.093, Table 2.

**Re-surgery because of complications.** Overall, 40 IBD patients (9.0% 30 CD, 10 UC) needed re-surgery because of postoperative complications.

Fourteen IBD patients (3.1%, 7 CD, 7 UC) needed re-surgery within 10 days after the initial surgery with no significant differences between exposed and un-exposed patients (CD, p = 1.000 and UC, p = 0.122, Table 2).

In 13 IBD patients (2.91%, 10 CD, 3 UC), re-surgery was necessary after 10 days and less than 3 months. Again, no significant differences were observed between patients exposed and unexposed patients (CD, p = 0.063 and UC, p = 1.000, Table 2).

**Univariable and multivariable analysis for any postoperative complication.** Multivariate logistic regression was performed for any minor postoperative complications, including

infections at the surgery site, wound healing disorders, intestinal paralysis, and elevation of inflammatory markers as well as for major complications, including anastomotic insufficiency, abscess formation, acute bleeding, peritonitis, and death (Table 3A–3E).

For CD and UC, important co-variates for logistic regression included age, disease duration, disease activity (remission/mild or moderate/severe), disease behavior according to the Montreal classification (14) and medical treatment at date of surgery, with biologic treatment (IFX, ADA, GOL, VDO, UST), plus TFO, and corticosteroid treatment, respectively.

Overall, biologic treatment within 12 weeks of surgery was not associated with an increased risk any complications in our cohort of 447 IBD patients who underwent resective surgical intervention (CD: p = 0.083 for minor complications, p = 0.633 for bleeding, peritonitis or death, and p = 0.106 for abscess formation or anastomotic insufficiency; UC: p = 0.151 for minor complications and p = 0.281 for major complications, Table 3A–3E).

For CD, higher age, and disease activity at date of surgery were found to be associated with occurrence of intestinal paralysis, nausea, and increased inflammatory parameters (p = 0.034 and p = 0.016, Table 3A). Furthermore, non-penetrating and non-stricturing disease behaviour, and disease duration were associated with the occurrence of major complications (p = 0.047 and p = 0.021, Table 3B).

For UC, corticosteroid treatment was associated with the occurrence of minor complications with nausea, intestinal paralysis, and increased inflammation parameters (p = 0.049, Table 3E).

No major complications were observed in UC patients.

## Discussion

In our retrospective observational study evaluating the postoperative outcome in 447 IBD patients with and without exposure to biologic treatment perioperatively, no increased risk of minor and major complications was observed, even when biologic treatment was given within 12 weeks before surgery (Tables 2 and 3A-3E), similar to the recently published PUCCINI trial [19], that also defined exposed patients as those having received biologics within 3 months before surgery.

The PUCCINI trial included 947 IBD patients with less than half of these receiving biologics before surgery (n = 382/947, 40.3%) whereas in our study cohort, 61.3% of patients received biologics (n = 275/447, Table 2), of which more than two thirds (189/275, 68.7%) received biologics even within 4 weeks prior to surgery, reflecting the pronounced disease activity in our patients and reflecting our strategy to operate patients within a tight time interval after the last infusions or injection of the biologics.

Our study also investigated immunomodulatory drugs other than anti-TNF-treatment, including vedolizumab, ustekinumab and tofacitinib (Tables 1, and 2).

For VDO and UST, data on perioperative treatment and postoperative outcome is still limited, because of the relatively recent approval of VDO and UST. However, some retrospective studies could position UST and VDO as relatively safe when used perioperatively [21–25].

For tofacitinib, a recently published retrospective review of all adult patients exposed to tofacitinib within 4 weeks of total abdominal colectomy concluded, that preoperative tofacitinib exposure in this setting may present an increased risk of postoperative venous thromboembolism (VTE) events [26]. Thus, prolonged VTE prophylaxis upon hospital discharge should be considered [26].

Regarding the perioperative use of TNF-alpha-antibodies, a large meta-analysis published in 2013, including a total of 18 studies with 4659 participants, controversially reported different results [12]. Studies limited to patients with CD demonstrated a statistically significant

increase of infectious (OR 1.93, 95% CI 1.28–2.89) and total (OR 2.19, 95% CI 1.69–2.84) complications and a trend towards an increase in of non-infectious complications (OR 1.73, 95% CI 0.94–3.17) whereas studies of patients with UC did not demonstrate significant increases of infectious (OR 1.39, 95% CI 0.56–3.45), non-infectious (OR 1.40, 95% CI 0.68–2.85), or total complications (OR 1.10, 95% CI 0.81–1.47) [12]. However, the authors argue that the small increase in apparent risk may well reflect the residual confounding factors rather than true cause and effect [12].

Other systematic reviews and meta-analyses also concluded that there might be an increased risk of postoperative infections when anti-TNF-treatment was given perioperatively [27–36].

However, other prospective and retrospective trials and systematic reviews with meta-analyses could not find an association between perioperative anti-TNF-treatment and an increased risk of postoperative complications [13–16].

Interestingly, the actual British Society of Gastroenterology consensus guidelines on the management on inflammatory bowel disease in adults from 2019 recommends discontinuation of anti-TNF-treatment, especially in CD, 6–8 weeks for IFX and 4 weeks for ADA before electively planned surgery [17], whereas the actual ECCO (European Crohn's and Colitis Organization) guidelines state that cessation of anti-TNF therapy, as well as vedolizumab or ustekinumab is not mandatory prior to surgery [18].

Our data, deriving from a tertiary referral center with rather sick patients, support the recommendations of the ECCO guidelines, showing no negative effect of direct exposure to biologic treatment prior to surgery even within 4 weeks of surgery.

With 97.1% of all patients (n = 443/447), surgery could be planned electively in our IBD patients. Only a minority of 8.95% (n = 40/447) needed re-surgery because of postoperative complications in our cohort (Table 2), with no significant differences in the subgroups with respect to exposure to biologics.

Two CD patients died in the postoperative period (Table 2), one exposed, one unexposed to biologics. When assessing this unfavorable postoperative outcome in the hospital's mortality rounds, it was agreed that both patients died due to their aggressive disease course.

In our cohort, multivariate logistic regression revealed the non-penetrating and non-stricturing disease behaviour B1, a long disease duration as well as age as independent risk factors in CD and in UC corticosteroid treatment as independent risk factor increasing the risk of postoperative complications. Biologic treatment did not increase this risk. It is unclear, why B1 disease behaviour is increasing the risk of postoperative complications in our cohort, it might be due to the small number of patients in this subcohort, but CD-patients with a B1 phenotype might be sicker than patients with a defined stricture or an abscess as seen in the B2/3 phenotypes.

Our data reflect the known risk of postoperative complications with perioperative corticosteroid treatment [18, 37–40] and as generally recommended in international guidelines, like the ECCO guidelines [41]. Cut-offs for increased surgical complications were observed between 10 mg and 40 mg prednisolone equivalent daily for more than 3–6 weeks [37–40]. Thus, corticosteroids should be tapered or stopped, whenever possible prior to surgery [17, 18].

We observed that CD patients with direct exposure to biologics within 4 weeks of abdominal surgery were more often undergoing minimal invasive surgery than CD patients receiving biologics longer before surgery (68.6%, n = 107/156 exposed within 4 weeks versus 48.3%, n = 28/58 exposed within 4–12 weeks before surgery, p = 0.020, Table 3). One explanation could be a better suppression of the inflammatory burden when biologics were given shortly before surgery, which might then influence the decision to perform minimal invasive surgery.

There are limitations of our study, mainly due to its retrospective character. In addition, no data on serum drug concentrations of biologic treatment were available. While the risk of complications seems to be associated with anti-TNF drug levels and IBD patients with active disease seem to have increased clearance of anti-TNF levels, our approach might potentially lead to a misclassification of exposure when defining the last use alone [19, 42, 43]. Also, no data on smoking status were available.

One strength of our study is the recruitment of a large number of patients from a single center resulting in a homogeneous patient cohort and data enrollment. A total of 447 IBD patients underwent abdominal surgery through a small number of only 7 senior surgeons, probably reducing confounding factors.

In conclusion, our actual single center study supports the results of prospective studies, e. g. the PUCCINI trial and the REMIND trial, which also conclude that biological treatment perioperatively is safe and does not increase the risk of postoperative complications in these patients. Corticosteroid treatment perioperatively was associated with minor complications in UC. Age, disease duration and disease behavior and activity, but not biological treatment, were associated with major complications in patients with CD.

## Supporting information

**S1 Data.**
(XLSX)

**S2 Data.**
(XLSX)

## Acknowledgments

We thank all participants from the Isarklinikum Munich and the MVZ VIVA Q Munich for their support to realize the study.

## Author Contributions

**Conceptualization:** Fabian Schnitzler, Franz Bader, Thomas Ochsenkühn.

**Data curation:** Fabian Schnitzler, Cornelia Tillack-Schreiber, Daniel Szokodi, Isabel Braun, June Tomelden, Maximilian Sohn, Franz Bader, Constanze Waggershauser, Thomas Ochsenkühn.

**Formal analysis:** Fabian Schnitzler, Cornelia Tillack-Schreiber, Daniel Szokodi, Isabel Braun, June Tomelden, Maximilian Sohn, Franz Bader, Constanze Waggershauser, Thomas Ochsenkühn.

**Funding acquisition:** Thomas Ochsenkühn.

**Investigation:** Fabian Schnitzler, Thomas Ochsenkühn.

**Methodology:** Fabian Schnitzler, Maximilian Sohn, Franz Bader, Constanze Waggershauser, Thomas Ochsenkühn.

**Project administration:** Fabian Schnitzler, Thomas Ochsenkühn.

**Resources:** Fabian Schnitzler, Cornelia Tillack-Schreiber, Daniel Szokodi, Isabel Braun, June Tomelden, Maximilian Sohn, Constanze Waggershauser.

**Supervision:** Franz Bader, Thomas Ochsenkühn.

**Validation:** Fabian Schnitzler, Maximilian Sohn, Franz Bader, Constanze Waggershauser, Thomas Ochsenkühn.

**Writing – original draft:** Fabian Schnitzler.

**Writing – review & editing:** Fabian Schnitzler, Franz Bader, Thomas Ochsenkühn.

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
