## [Decision Letter · Decision Letter 0]

13 Jun 2023

PONE-D-23-13172Safety of perioperative treatment with biologics in patients with inflammatory bowel disease undergoing bowel surgery: experience from a large urban centerPLOS ONE

Dear Dr. Schnitzler,

Thank you for submitting your manuscript to PLOS ONE. After careful consideration, we feel that it has merit but does not fully meet PLOS ONE’s publication criteria as it currently stands. Therefore, we invite you to submit a revised version of the manuscript that addresses the points raised during the review process.

We look forward to receiving your revised manuscript.

Kind regards,

Marco Clementi, Assistant Professor

Academic Editor

PLOS ONE

Journal Requirements:

"This work was financially supported by the European Crohn’s and Colitis Foundation

(ECCS) and HEXAL AG"

"NO authors have competing interests"

Additional Editor Comments:

Minor revisions are requested

Reviewers' comments:

Reviewer's Responses to Questions

**Comments to the Author**

1. Is the manuscript technically sound, and do the data support the conclusions?

Reviewer #1: Yes

Reviewer #2: Yes

2. Has the statistical analysis been performed appropriately and rigorously? 

Reviewer #1: Yes

Reviewer #2: Yes

3. Have the authors made all data underlying the findings in their manuscript fully available?

Reviewer #1: Yes

Reviewer #2: Yes

4. Is the manuscript presented in an intelligible fashion and written in standard English?

Reviewer #1: Yes

Reviewer #2: No

5. Review Comments to the Author

Reviewer #1: Many hanks to the authors for giving me the opportunity to revise this article. This is a retrospective analysis about the treatment with biologics before surgery in IBD patients.

Some comments:

- why postoperative complications were not classify according to Clavien-Dindo classification?

Reviewer #2: dear Authors, thanks for the opportunity to review your work. I congratulate with you all for the well conducted and interesting investigation. Herebelow my comments:

Intestinal paralysis meant prolonged postoperative ileus? for how many days before flatus or oral food intake? please clarify

Nausea meant a symptom associated with the above? or do you mean incidence of PONV?

Insufficiency means anastomotic leak? subclinical or overt disruption? please give details and clarify

CD patients showed almost threefold increase in anastomotic leak rate compared to UC (17 vs 6). was it statistically significant? please make a comment on this.

Some sentences are a bit complicate and confusing. please make a full evaluation by a motherlanguage.

6. PLOS authors have the option to publish the peer review history of their article (what does this mean?). If published, this will include your full peer review and any attached files.

Reviewer #1: No

Reviewer #2: No

---

## [Author Response · Author response to Decision Letter 0]

4 Aug 2023

Dear Professor Clementi,

Thank you for consideration of our Manuscript PONE-D-23-13172 “Safety of perioperative treatment with biologics in patients with inflammatory bowel disease undergoing bowel surgery: experience from a large urban center”. 

Please find attached below the point-by-point answers to the comments of the reviewer of our manuscript, as well as the manuscript with corrections and a clean copy version. 

We hope that our actual manuscript with the corrections now fulfills the requirements of Plos One. 

Thank you for re-consideration of our corrected manuscript for publication to Plos One. 

With kindest regards,

Fabian Schnitzler

We thank the Editors for this important comment. The citation style was now adapted to the PLOS ONE style templates. 

"This work was financially supported by the European Crohn’s and Colitis Foundation

(ECCS) and HEXAL AG"

We thank the Editors for this comment. The funders had no role in our study. Hence, now we included the statement “The funders had no role in study design, data collection and analysis, decision to publish, or preparation of the manuscript.” in the paragraph “Funding” of our manuscript. 

"NO authors have competing interests"

We completed the Competing Interests of the authors now on the online submission form for all authors as follows:

• T. Ochsenkühn has received travel, research and study grants or honoraria for consultations or lectures from AbbVie, Amgen, Biogen, Celltrion, Fresenius, Hexal, Janssen, Lilly, MSD, Viatris, Vifor, Pfizer and Takeda.

• F. Schnitzler has received travel grants and honoraria for consultations or lectures from Abbvie, Amgen, Biogen, Hexal, Janssen, Lilly, MSD, Viatris, Pfizer and Takeda.

• C. Tillack has received travel grants or honoraria for consultations from Janssen, Galapagos and Lilly. 

• D. Szokodi has received travel grants or honoraria for consultations or lectures from AbbVie, Amgen, Biogen, BMS, Celltrion, Shire, Lilly, Janssen, Pfizer and Takeda.

• M. Sohn has received honoraria for lectures from Galapagos, Lilly, and Taked.

• F. Bader has received travel, research and study grants or honoraria for consultations or lectures from Intuitive Surgical Medtronic, MTIGER and Takeda

• C. Waggershauser has received travel grands and lecture fees from Biogen, Janssen and Lilly. 

• This did not alter the authors' adherence to all the PLOS ONE policies on sharing data and materials, as detailed online in the guide for authors.

• The other authors have no conflicts of interest to disclose.

• No writing assistance was utilized in the production of this manuscript.

• There are no patents, products in development or marketed products to declare.

Thank you for this important comment. We again discussed this point with the statisticians. There was a misunderstanding. All relevant data for the paper are already included in the tables of the manuscript. Even more, all data are included for future meta-analyses. We updated this information in the submission site: “Yes, all data are fully available without restrictions.” And “All relevant data are within the manuscript and its Supporting Information files.” Now we uploaded the data of the manuscript as supporting information files. 

We again reviewed our reference list. It is complete and correct. Thank you for this comment. 

Additional Editor Comments:

Minor revisions are requested

Reviewers' comments:

Reviewer's Responses to Questions

Comments to the Author

1. Is the manuscript technically sound, and do the data support the conclusions?

Reviewer #1: Yes

Reviewer #2: Yes

2. Has the statistical analysis been performed appropriately and rigorously?

Reviewer #1: Yes

Reviewer #2: Yes

3. Have the authors made all data underlying the findings in their manuscript fully available?

Reviewer #1: Yes

Reviewer #2: Yes

4. Is the manuscript presented in an intelligible fashion and written in standard English?

Reviewer #1: Yes

Reviewer #2: No

5. Review Comments to the Author

Reviewer #1: Many thanks to the authors for giving me the opportunity to revise this article. This is a retrospective analysis about the treatment with biologics before surgery in IBD patients.

Some comments:

- why postoperative complications were not classify according to Clavien-Dindo classification?

Thank you for this important comment. We did not classify the postoperative outcomes according to the Clavien-Dindo classification according to its retrospective study design. We discussed this point with our surgeons. We could not identify hundred percent of information needed to fulfill the Clavien-Dindo classification. Hence, the classification would not be as precise as needed. Based on the large prospective PUCCINI trial (Cohen BL et al. Gastroenterology 2022; 163:204–221) we choose a descriptive classification best depicting our available data from the patients’ charts. 

Reviewer #2: dear Authors, thanks for the opportunity to review your work. I congratulate with you all for the well conducted and interesting investigation. Here below my comments:

Intestinal paralysis meant prolonged postoperative ileus? for how many days before flatus or oral food intake? please clarify

Intestinal paralysis means a prolonged postoperative ileus and is defined as a dysmotility of the small bowel and the colon for more than 72 hours in combination with nausea, and/or vomiting, abdominal pain, and abdominal distension. This is now clarified in the actual version of the paper in the paragraph “End points and statistical analysis” on pages 7 and 8 :” Intestinal paralysis was seen as a prolonged postoperative ileus and is defined as a dysmotility of the small bowel and the colon for more than 72 hours in combination with nausea, and/or vomiting, abdominal pain, and abdominal distension.”

Nausea meant a symptom associated with the above? or do you mean incidence of PONV?

Nausea meant a symptom associated with the above and not PONV.

Insufficiency means anastomotic leak? subclinical or overt disruption? please give details and clarify

Anastomotic insufficiency meant anastomotic leackage. Postoperative intraabdominell abscess, fistula, peritonitis, and anastomotic leackage are subsumed as postoperative intrabdominal septic complications (IASC).

Thereof, we selected anastomotic leackage for more detailed information on postoperative complications, which is defined as overt disruption of the stapler or suture line, identified by endoscopic or surgical exploration.

This is now clearly mentioned in the actucal version of the paper, in the paragraph “End points and statistical analysis” on page 8: “Anastomotic insufficiency was seen as anastomotic leakage and defined as overt disruption of the stapler or suture line, identified by endoscopic or surgical exploration”.

CD patients showed almost threefold increase in anastomotic leak rate compared to UC (17 vs 6). was it statistically significant? please make a comment on this.

In a total of 17 of 334 CD patients occurred anastomotic leakage (5.09%) compared to a total of 6 UC patients in the UC cohort of 113 patients (5.31%). This was not statistically significant (p= 1.000, chi square test). This is now mentioned in the paper on page 15, in the paragraph “Major complications”: No significant difference was observed in CD patients versus UC patients regarding anastomotic insufficiency (CD, 17/334 patients, 5.09%, vs. UC, 6/113 patients, 5.31%, p=1.000, table 2).

Some sentences are a bit complicate and confusing. please make a full evaluation by a mother language.

We asked a native English speaker to read the final version of our manuscript and to make corrections. Thank you for the comment. 

6. PLOS authors have the option to publish the peer review history of their article (what does this mean?). If published, this will include your full peer review and any attached files.

Do you want your identity to be public for this peer review? For information about this choice, including consent withdrawal, please see our Privacy Policy.

Reviewer #1: No

Reviewer #2: No

---

## [Editor Report · Decision Letter 1]

16 Aug 2023

Safety of perioperative treatment with biologics in patients with inflammatory bowel disease undergoing bowel surgery: experience from a large urban center

PONE-D-23-13172R1

Dear Dr. Dr. Fabian Schnitzler,

We’re pleased to inform you that your manuscript has been judged scientifically suitable for publication and will be formally accepted for publication once it meets all outstanding technical requirements.

Kind regards,

Marco Clementi, Assistant Professor

Academic Editor

PLOS ONE
---

## [Editor Report · Acceptance letter]

29 Aug 2023

PONE-D-23-13172R1 

Safety of perioperative treatment with biologics in patients with inflammatory bowel disease undergoing bowel surgery: experience from a large urban center 

Dear Dr. Schnitzler:

I'm pleased to inform you that your manuscript has been deemed suitable for publication in PLOS ONE. Congratulations! Your manuscript is now with our production department. 

Kind regards, 

on behalf of

Dr. Marco Clementi 

Academic Editor

PLOS ONE